# Associations between Public Transport Accessibility around Homes and Schools and Walking and Cycling among Adolescents

**DOI:** 10.3390/children7040030

**Published:** 2020-04-06

**Authors:** Syafiqah Hannah Binte Zulkefli, Alison Barr, Ankur Singh, Alison Carver, Suzanne Mavoa, Jan Scheurer, Hannah Badland, Rebecca Bentley

**Affiliations:** 1Centre for Health Equity, Melbourne School of Population and Global Health, Melbourne, VIC 3010, Australia; syafiqah.z95@gmail.com (S.H.B.Z.); alison.barr@unimelb.edu.au (A.B.); suzanne.mavoa@unimelb.edu.au (S.M.); brj@unimelb.edu.au (R.B.); 2Mary MacKillop Institute for Health Research, Australian Catholic University, Melbourne, VIC 3000, Australia; Alison.Carver@acu.edu.au; 3Centre for Urban Research, Royal Melbourne Institute of Technology (RMIT), Melbourne, VIC 3000, Australia; jan.scheurer@rmit.edu.au (J.S.); hannah.badland@rmit.edu.au (H.B.); 4Centre for Epidemiology and Biostatistics, Melbourne School of Population and Global Health, Melbourne, VIC 3010, Australia

**Keywords:** adolescents, public transport accessibility, physical activity, household travel survey

## Abstract

Good public transport accessibility is associated with active travel, but this is under-researched among adolescents. We tested associations between public transport accessibility and active travel among school-going adolescents (12–18 years; *n* = 1329) from Melbourne, Australia analysing Victorian Integrated Survey of Travel and Activity data. Outcomes included main mode of transport to school and accumulating ≥20 min of active travel over the day. Low and high compared to no public transport accessibility around homes were associated with higher odds of public transport use (low (odds ratio (OR): 1.94 95% confidence interval (CI): 1.28, 2.94) high (OR: 2.86 95% CI: 1.80, 4.53)). Low and high public transport accessibility around homes were also associated with higher prevalence of achieving ≥20 min of active travel (low (prevalence ratio (PR): 1.14 95% CI: 0.97, 1.34) high (PR: 1.31 95% CI: 1.11, 1.54)) compared to none. Public transport accessibility around schools was associated with public transport use (low (OR: 2.13 95% CI: 1.40, 3.24) high (OR: 5.07 95% CI: 3.35, 7.67)) and achieving ≥20 min of active travel (low (PR: 1.18 95% CI: 1.00, 1.38) high (PR: 1.64 95% CI: 1.41, 1.90)). Positive associations were confirmed between public transport accessibility and both outcomes of active travel.

## 1. Introduction

The health benefits of physical activity in preventing life-threatening chronic diseases including cardiovascular diseases, obesity, diabetes and cancer is well established [1,2,3,4,5,6]. Separate health guidelines exist for physical activity for children and adolescents compared to adults. Children and adolescents must accumulate at least an hour of moderate- to vigorous-intensity physical activity every day for health benefits [7]. However, adolescents spend less time in physical activity compared with children and often fail to meet recommended minimum levels. Only one in ten adolescents meet recommendations of 60 min of daily activity compared with one in three children aged less than 13 years in Australia [7].

Rising levels of obesity and cardiovascular disease in the population [8] require rigorous policy responses to increase physical activity among both children and adolescents. Adolescents have the lowest rate of active travel (walking/cycling for transport) of any age group regardless of total household income and ethnicity [9,10], so active travel represents a strategic opportunity to increase physical activity in this group. Walkable environments, as measured by indices that include dwelling density, connectivity and land use mix for example, are positively associated with walking and active travel in a range of populations [11,12], including adolescents [13], so they represent a potential focus for intervention. It is important, therefore, to understand the extent to which the built environment around homes and schools support walking and cycling among adults and children [13,14,15,16]. The journey to school for school-going adolescents (12–18 years of age) is identified as a key opportunity to increase their physical activity [14]. Additionally, there is strong evidence that health promoting behaviours established in adolescence track into adulthood [17,18,19]. 

Many studies [13,14,15,16] investigate the association between the residential- and neighbourhood-level built environment and active travel. These often do not account for all built environments experienced by people through the day (such as work and school), leading to the residential effect fallacy [20]. This preoccupation with the local home neighbourhood has also been to the detriment of research on built environment characteristics that support regional travel patterns, for example the commute to work or school. The journey to school is perhaps the most important trip for adolescents, as it offers habitual opportunity for incidental physical activity. An important consideration, however, is that adolescents typically travel further to secondary schools than children do to primary schools, since catchments tend to be wider, and therefore many adolescents do not live within walking distance of their school [21]. Consequently, they are more dependent on a regional commute by cycle, public transport or private car [22].

Along with cycling infrastructure, public transport accessibility is a characteristic of the built environment that is increasingly relevant to policy and research on health-promoting built environments [23,24,25,26,27]. Public transport is often regarded as an active mode of travel as there is usually a walk required at one or more parts of the journey [28,29]. Therefore, public transport infrastructure is arguably an essential component of walking infrastructure that should be designed to efficiently connect walkable neighbourhoods across the city. Public transport accessibility measures capture these opportunities for active travel through vehicle access and egress legs on longer regional journeys, such as the journey from home to secondary school or work [27,28,29,30,31]. Research findings support the association between public transport use and walking—on average, public transport users walk more than private car users and sometimes as much or more than those using only active modes [27,28,31]. Modelling by Ermagun and Levinson [26] demonstrated that a decrease in public transport accessibility could actually lead to a decrease in physical activity among secondary students. However, there is a paucity of research exploring the association between public transport accessibility and adolescents’ active travel to school.

This study aims to investigate if the public transport accessibility of the home and school neighbourhoods are associated with a) walking/cycling or taking public transport to school and b) accumulating 20 min or more of daily active travel (to school and/or other destinations) in adolescents aged 12 to 18 years.

## 2. Materials and Methods

### 2.1. Study Population

The sample for this cross-sectional study was selected from the Victorian Survey of Travel and Activity (VISTA) 2012–2016, a cross-sectional household travel survey of Greater Melbourne and Geelong in the state of Victoria, Australia, described elsewhere [32]. VISTA runs continuously throughout the year, surveying different households on a single day. Questionnaires were sent to homes within a random selection of Mesh Blocks (the smallest spatial area defined by the Australian Bureau of Statistics (ABS) [33] and used for census data collection). All members of each household were asked to complete a questionnaire of personal and household information, as well as a travel diary documenting all travel on the specified survey day. Travel data included trip origins and destinations, and the amount of time spent on each trip undertaken on the survey day by travel mode (car, walking, bus, etc.). For confidentiality purposes, home addresses were randomised to within 100 m of the actual address, and all trip origins and destinations were geocoded using Geographic Information Systems (GIS) software.

A total of 46,562 people responded to the VISTA 2012–2016 survey of which, 3735 were adolescents aged 12–18 years. For the purposes of analysis, we excluded people who did not take a journey to education (*n* = 2087), who were not attending secondary school (*n* = 294), who travelled to education on a weekend day (*n* = 23), whose distance to school could not be calculated (*n* = 1), and who travelled to a place of education for purposes other than schooling (*n* = 4). Adolescents aged 19 and above transitioning from school to university or employment were excluded from this analysis.

Data from a total of 1326 secondary students who journeyed to secondary school on a school day were finally analysed. 

### 2.2. Outcomes

The first outcome measure, the main mode of transport to school, was split into three categories based on the main transportation mode reported for the morning journey from home to school. The outcome was categorised as (1) private motorised transport (car, school bus, taxi or other) (2) walking or cycling and (3) public transport, and the reference group defined as private motorised transport users. This categorisation was selected to account for 1) the high number of respondents taking multi modal journeys to school, and 2) the fact the average time spent in active travel was similar in active travel users and public transport users, as shown in Figure 1. The main mode was determined by the maximum time spent in a single mode of transport on the journey to school.

The second outcome, meeting 20 min of active travel, was calculated by adding the total amount of time spent in active travel across the survey day. This included time spent walking or cycling to and from public transport. While the physical activity guidelines recommend 60 min of physical activity per day [34,35,36], few in this study met this recommendation only through active travel. Therefore, participants were divided into two groups—those who accumulated ≥20 min of active travel and those who did not, as this amount of physical activity contributes a third of daily requirements.

### 2.3. Exposures

#### Public Transport Measure

The exposure measure for public transport accessibility was derived from Spatial Network Analysis for Multi-modal Urban Transport Systems (SNAMUTS) data [37]. SNAMUTS measures public transport accessibility using a network analysis approach and generates different indicators of how effective the public transport system is [38]. These include indices such as the centrality of the network, the overall connectivity of a stop to the entire public transport system across the city, and whether the system can support current demand and potential growth. This means that SNAMUTS measures the individual’s ease of accessibility to destinations across the whole public transport network, rather than just their access to the closest public transport stop, making it a more sophisticated measure than those other studies have used [25,27]. This analysis used the SNAMUTS composite index which combines several other SNAMUTS measures to present overall accessibility scores of the whole public transport system from a single location. The composite index score for the Mesh Block of the home and school address were classified into none (did not meet a minimum service frequency standard), low (below mean public transport accessibility), and high (equal and above-mean public transport accessibility). 

### 2.4. Covariates

Covariates included gender (male/female), age (in years), household level weekly income (in Australian dollars, quintiles), household size (number of people), vehicle ownership (≤1 car, 2+ cars), neighbourhood socioeconomic status (using the Australian Bureau of Statistics measure of Socioeconomic Indices for Areas (SEIFA) Index of Relative Disadvantage (IRSD) categorised into quartiles), distance traversed between home and school (calculated as the shortest network path along the road system computed by Geographic Information Systems and categorised into categories based on tertiles “<2.5 km”, “2.5–8.5 km”, and “≥8.5 km”), as well as household and school walkability (measured using a Walkability Index which scored Mesh Blocks according to a combination of residential density, land mix use, and street connectivity).

### 2.5. Statistical Analysis

Multinomial regression models were fitted to test the associations between the built environment characteristics and adolescents’ main mode of transport. Odds ratios (OR) with 95% confidence intervals (CI) comparing two categories (active travel and public transport) with a reference group (motorised vehicle use) were estimated using this modelling approach to quantify the strength of association and its uncertainty. Modified Poisson models with robust standard errors were used to estimate the unadjusted and adjusted prevalence ratios of achieving 20 min or more of activity per day. Estimates obtained from regression models adjusted for covariates account for key confounding factors (gender, age, household weekly income, household size, vehicle ownership, neighbourhood socioeconomic status, distance between home and school, household walkability and school walkability) in the relationships between public transport accessibility and active travel outcomes than unadjusted models. All models accounted for clustering of the outcomes within SA1s (geographical areas made of whole Mesh Blocks) [33] by use of a multilevel modelling technique. While clustering was present in multinomial models, it was absent in Poisson models.

Spatial analyses were conducted using ArcGIS 10.4 (ESRI, 2016, Redlands, CA, USA) Geographic Information Systems software. Statistical analyses were conducted using Stata/MP 15.1 (StataCorp, College Station, TX, USA). Ethics approval was granted by the University of Melbourne Human Research Ethics Committee (ID: 1442864.1).

## 3. Results

Of the 1329 adolescents included in this analysis, 687 (52%) of the participants were males (Table 1). The median age was 15 years, with an interquartile range of 13 to 16 years. Nearly 64% had their travel diaries completed by proxy. There was no incomplete reporting of travel. Most participants (65%) travelled by private motorised transport to school followed by the more active modes of walking/cycling (18%) and public transport (17%). More than half (57%) did not accumulate 20 min of daily active travel.

### 3.1. Outcome: Main Mode of Transport to School

In unadjusted models, public transport accessibility around homes was associated with walking or cycling and the use of public transport to school (Table 2). Any public transport accessibility around homes was associated with greater odds of walking or cycling (OR = 1.74; 95% Confidence Intervals (CI) = 1.17, 2.57 for low and OR = 1.93; 95% CI = 1.27, 2.92 for high). The odds of using public transport was also considerably higher among those with any transport accessibility around homes (OR = 2.49; 95% CI = 1.71, 3.63 for low and OR = 3.81; 95% CI = 2.63, 5.52 for high). However, adjustment for covariates explained the association between public transport accessibility around homes and walking or cycling to school. In contrast, there was only marginal attenuation in the association between public transport accessibility around homes and the use of public transport (OR = 1.94; 95% CI = 1.28, 2.94 for low and OR = 2.86; 95% CI = 1.80, 4.53 for high) in fully adjusted models (Table 2).

Similarly, for public transport accessibility around schools, in fully adjusted models, associations were only observed for public transport as a main mode of transport to school (OR = 2.13; 95% CI = 1.40, 3.24 for low and OR = 5.07; 95% CI = 3.35, 7.67 for high) (Table 3).

### 3.2. Outcome: 20 Min of Active Travel Daily

In both unadjusted and adjusted models, the likelihood of accruing 20 min or more of active travel across the day was positively associated with public transport accessibility of adolescents’ home and school environments. Upon adjustment, the prevalence ratio of accruing 20 min or more of active travel for those with any public transport accessibility around homes was higher (PR = 1.14; 95% CI = 0.97, 1.34 for low and PR = 1.31; 95% CI = 1.11, 1.54 for high) than those without any accessibility (Table 2). Similarly, the adjusted prevalence ratio of accruing 20 min or more of active travel for those with any public transport accessibility around schools was higher (PR = 1.18; 95% CI = 1.00, 1.38 for low and PR = 1.64; 95% CI = 1.41, 1.90 for high) than those without any accessibility (Table 3).

## 4. Discussion

In this study, increased public transport accessibility at both the home and school environment was associated with adolescents using public transport as the main mode of travel on the journey to school. Increased public transport accessibility at both localities was also found to be associated with accruing 20 min or more of active travel across the day. Expectedly, the associations between public transport accessibility at both home and school, and walking and cycling, was explained by confounding in this group of adolescents. This may be because as public transport accessibility increases, so too does transit use, resulting in fewer adolescents travelling to school by active travel mode only.

Better public transport accessibility around schools was associated with an increased likelihood of adolescents accruing 20 or more minutes of active travel per day. The school environment appeared to be just as, if not more, important than the home environment in predicting public transport use and active travel. This suggests that the residential effect fallacy may be at work in studies of home neighbourhoods, and that we need to undertake research on the relationship between the built environment and travel modes in multiple activity spaces. This is especially pertinent for understanding environmental influences on adolescents’ active travel patterns, as they extend from a childhood focus on the home neighbourhood into a more adult territorial range [20]. It may also be that adolescents have more discretion in and reliance on active modes of travel when they are closer to the school environment, and away from opportunities for private vehicular travel provided by parents at home. 

### Strengths and Limitations

To our knowledge, this is the only study that has examined public transport accessibility at both home and school environments and its association with adolescents’ active travel. Investigating the primary and secondary activity spaces of home and school is a key strength of this study as it addresses residential effect fallacy [20].

Another strength of this study is that SNAMUTS is a superior measure of public transport accessibility compared to those used in most previous research. Measures such as service frequency at the nearest stop have limited relevance, as they do not fully encompass accessibility to all destinations from the point of origin. However, by including factors such as centrality to all locations of interest, public transport speed and service frequency, land use intensity, directness of journeys, flexibility of transfers and integration of multimodal public transport services, etc., SNAMUTS allows a more holistic view of accessibility to a range of potential destinations through the public transport system.

VISTA included detailed demographic information on both an individual and household level and comprehensive information on individuals’ travel behaviours, such as the length of travel by different modes in the same journey. This allowed for multiple relevant confounding factors to be accounted for in the analysis of association exposures and outcomes.

This study has several limitations. Firstly, it was a cross-sectional study design and therefore cannot generate causal evidence as temporality between exposures and outcomes could not be established. Secondly, household travel survey data reported on a single day may not reflect habitual travel behaviour and may be prone to inaccuracies in reporting. Additionally, it could not be established whether any changes in physical activity were due to the environment, or whether a person’s proclivity for physical activity led to their environment. This could have been mitigated by accounting for self-selection into residential areas based on household travel preferences [39], however, this information was not available. 

## 5. Conclusions

This study provides evidence that adolescents with higher public transport accessibility around homes and schools are more likely to use public transport and accumulate more minutes of active travel. Thus, investments in improving public transport infrastructure and services may also represent an important opportunity to increase adolescents’ physical activity. This study is of particular importance as it focuses on adolescents, a population that is often overlooked and understudied. From a life-course perspective, adolescents are significantly increasing their territorial range and their needs and capacity for regional travel, but are as yet too young to be licensed to drive, making them an ideal target group for interventions seeking to increase healthy and sustainable travel modes such as public transport. 

As health promotion efforts to increase physical activity in this age group have had limited effect, future policy around urban environments should consider public transport accessibility when considering the health of our adolescent population. It is important that we continue to design environments that encourage active travel amongst adolescents, particularly around destinations, setting them up for a healthier future.

## Figures and Tables

**Figure 1 children-07-00030-f001:**
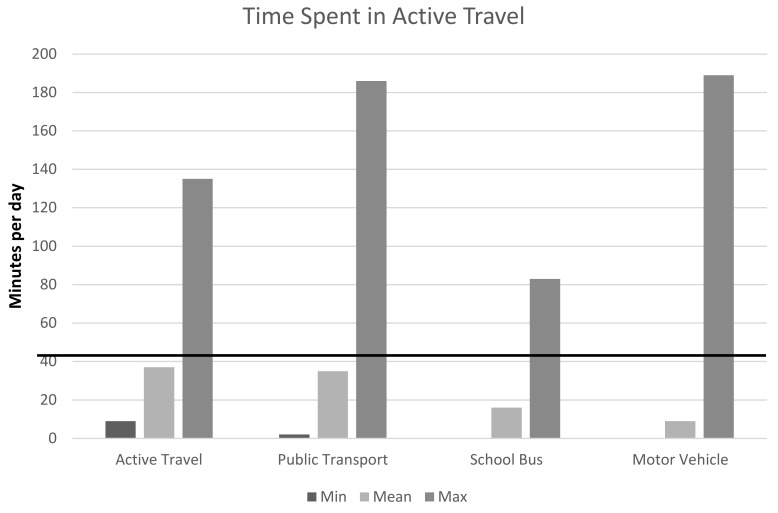
Total time spent in active travel by main mode of transportation to school.

**Table 1 children-07-00030-t001:** Sample characteristics (*n* = 1329).

Variable Name	Categories	*n*	%
Age of respondents (years)	12	145	10.9
	13	211	15.9
	14	238	17.9
	15	237	17.8
	16	233	17.5
	17	188	14.2
	18	77	5.8
Gender	Male	687	51.7
	Female	642	48.3
Distance Travelled from Home to School	<2.5 km	331	24.9
	2.5–8.5 km	665	50.0
	≥8.5 km	333	25.1
Usual number of residents in household	2	48	3.6
	3	225	16.9
	4	570	42.9
	5	344	25.9
	6	98	7.4
	7	26	2.0
	8	10	0.8
	9	8	0.6
Household Income Groups	$0–$799	180	13.5
	$800–$1249	175	13.2
	$1250–$1999	302	22.7
	$2000–$2999	333	25.1
	$3000+	339	25.5
Total Vehicles Owned	0 or 1	300	22.6
	2 or more	1029	77.4
Neighbourhood Socioeconomic Status	Quartile 1 (Lowest)	333	25.1
	Quartile 2	333	25.1
	Quartile 3	333	25.1
	Quartile 4 (Highest)	330	24.8
How was diary completed?	Proxy-Reported	846	63.8
	Self-Reported	480	36.2
Main Mode of Transport to School	Private Motorised	857	64.5
	Public Transport	228	17.2
	Active Travel	244	18.4
Accrued 20 Minutes+ of Active Transport per Day	No	754	56.9
	Yes	572	43.1
Household Walkability	Tertile 1 (Low)	447	33.6
	Tertile 2 (Medium)	577	43.4
	Tertile 3 (High)	305	23.0
School Walkability	Tertile 1 (Low)	447	33.6
	Tertile 2 (Medium)	441	33.2
	Tertile 3 (High)	441	33.2
Household Public Transport Accessibility	None	921	69.3
	Low (below mean)	212	16.0
	High (above mean)	196	14.8
School Public Transport Accessibility	None	738	55.5
	Low (below mean)	306	23.0
	High (above mean)	285	21.4

**Table 2 children-07-00030-t002:** Associations between public transport accessibility at home on the likelihood of taking active or public transport to school compared to cars, and the likelihood of achieving 20 min of active travel on the journey to school (*n* = 1329).

	Unadjusted		Adjusted	
Travel Mode	OR	95% CI	OR	95% CI
Active Travel				
None	1.00		1.00	
Low	1.74	(1.17, 2.57)	1.09	(0.66, 1.82)
High	1.93	(1.27, 2.92)	1.00	(0.56, 1.78)
Public Transport				
None	1.00		1.00	
Low	2.49	(1.71, 3.63)	1.94	(1.28, 2.94)
High	3.81	(2.63, 5.52)	2.86	(1.80, 4.53)
20+ minutes active travel	PR	95% CI	PR	95% CI
None	1.00		1.00	
Low	1.33	(1.14, 1.56)	1.14	(0.97, 1.34)
High	1.67	(1.46, 1.91)	1.31	(1.11, 1.54)

Adjusted for gender, age, household weekly income, household size, vehicle ownership, neighbourhood socioeconomic status, distance between home and school, household walkability and school walkability.

**Table 3 children-07-00030-t003:** Associations between public transport accessibility at school on the likelihood of taking active or public transport to school compared to cars, and the likelihood of achieving 20 min of active travel on the journey to school (*n* = 1329).

	Unadjusted		Adjusted	
Travel Mode	OR	95% CI	OR	95% CI
Active Travel				
None	1.00		1.00	
Low	1.40	(0.99, 1.98)	1.57	(0.98, 2.51)
High	1.20	(0.80, 1.81)	1.44	(0.82, 2.52)
Public Transport				
None	1.00		1.00	
Low	2.33	(1.59, 3.42)	2.13	(1.40, 3.24)
High	6.40	(4.51, 9.10)	5.07	(3.35, 7.67)
20+ minutes active travel	PR	95% CI	PR	95% CI
None	1.00		1.00	
Low	1.21	(1.03, 1.42)	1.18	(1.00, 1.38)
High	1.73	(1.52, 1.97)	1.64	(1.41, 1.90)

Adjusted for gender, age, household weekly income, household size, vehicle ownership, neighbourhood socioeconomic status, distance between home and school, household walkability and school walkability.

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
