# Peer review of "Associations between Public Transport Accessibility around Homes and Schools and Walking and Cycling among Adolescents"

_children, 2020, doi:10.3390/children7040030_

Round 1
Reviewer 1 Report
This is an interesting paper with policy relevant findings. The analysis appears sound however it should be noted that this reviewer is not an expert in quantitative modelling of the type reported or, specifically, application of the SNAMUTS tool. The paper should be reviewed by experts with these skills.
Specific comments are as follows:
The paper requires a thorough edit – as such there are many errors and places where the wording is clumsy and points are repeated. There is an unfinished sentence in the abstract (L37/8).
"Adolescents 45 must accumulate at least an hour of moderate to vigorous intensity physical activity every day for 46 health benefits [7]" - this is the guideline for children and adolescents and this sentence should not single out adolescents specifically.
Given the Journal’s focus I assume this is obvious to most of its readership however coming from a transport background it would be nice to clarify up front the exact age group of adolescent.
The paragraph beginning "Along with cycling infrastructure, public transport accessibility is a characteristic of the built environment that is increasingly relevant..." contains a lot of repetition and at times doesn’t make a lot of sense. It is the paragraph establishing the research gap and needs to be given greater consideration.
“This study aims to investigate if the public transport accessibility of the home and school..” should be “This study aims to investigate if access to public transport from the home or school environment…”
The description of the VISTA study and data is poorly written, almost implying that VISTA is conducted for this study alone. See McCarthy, L., Delbosc, A., Currie, G., Molloy, A.: Parenthood and cars: a weakening relationship? Transportation 4, 1–19 (2018) for some tips on how to better introduce VISTA.
In the context of the preceding paragraph describing mode to school, the description of the active travel per day needs to be clarified as taken from the respondent’s entire day of transport rather than just the mode to school, assuming that it is.
Page 7, there are two rouge sentences: "Adjusted for sex, age, household weekly income. Household size, vehicle ownership, neighbourhood socioeconomic status, distance between home and school, household walkability and school walkability" - not sure where they belong.
Author Response
Comment 1: This is an interesting paper with policy relevant findings. The analysis appears sound however it should be noted that this reviewer is not an expert in quantitative modelling of the type reported or, specifically, application of the SNAMUTS tool. The paper should be reviewed by experts with these skills.
Response: Thank you for acknowledging our contribution.
Specific comments are as follows:
Comment 2: The paper requires a thorough edit – as such there are many errors and places where the wording is clumsy and points are repeated. There is an unfinished sentence in the abstract (L37/8).
Response: We have conducted a thorough review and edited where required to improve readability. Thank you for pointing out the unfinished sentence which has now been deleted.
Comment 3: "Adolescents 45 must accumulate at least an hour of moderate to vigorous intensity physical activity every day for 46 health benefits [7]" - this is the guideline for children and adolescents and this sentence should not single out adolescents specifically.
Response: The revised sentence (lines 44-46) now states:
“Children and adolescents must accumulate at least an hour of moderate to vigorous intensity physical activity every day for health benefits [7].”
Comment 4: Given the Journal’s focus I assume this is obvious to most of its readership however coming from a transport background it would be nice to clarify up front the exact age group of adolescent.
Response: Our study sample comprises of school-going adolescents aged 12-18 years, as described at lines 52, 88 and 102. However, there is no universal definition of the age-group of adolescents given that adolescence is a ‘transitional phase of growth and development between childhood and adulthood’ (ref https://www.britannica.com/science/adolescence). For example the World Health Organisation considers adolescents to be aged from 10-19 years, but the Australian Government’s Dept of Health has different physical activity guidelines for children and youth (5-17 years) and for adults (aged 18+ years) (https://www.health.gov.au/health-topics/exercise-and-physical-activity). .
Comment 5: The paragraph beginning "Along with cycling infrastructure, public transport accessibility is a characteristic of the built environment that is increasingly relevant..." contains a lot of repetition and at times doesn’t make a lot of sense. It is the paragraph establishing the research gap and needs to be given greater consideration.
Response: We have edited the paragraph beginning at line 80 (revised version) as follows:
“Along with cycling infrastructure, public transport accessibility is a characteristic of the built environment that is increasingly relevant to policy and research on health-promoting built environments [24-28]. Public transport is often regarded as an active mode of travel as there is usually a walk required at one or more parts of the journey [29,30]. Therefore, public transport infrastructure is arguably an essential component of walking infrastructure that should be designed to efficiently connect walkable neighbourhoods across the city. Public transport accessibility measures capture these opportunities for active travel through vehicle access and egress legs on longer regional journeys, such as the journey from home to secondary school or work [28-32]. Research findings support the association between public transport use and walking: on average public transport users walk more than private car users and sometimes as much or more than those using only active modes [28,29,32]. Modelling by Ermagun and Levinson [27] demonstrated that a decrease in public transport accessibility could actually lead to a decrease in physical activity among secondary students. However, there is a paucity of research exploring the association between public transport accessibility and adolescents’ active travel to school.”
Comment 6: “This study aims to investigate if the public transport accessibility of the home and school..” should be “This study aims to investigate if access to public transport from the home or school environment…”
Response: We have not made the suggested change as we are referring to the accessibility of public transport within these neigbourhoods (for journeys to, from and within these neigbourhoods).
Comment 7: The description of the VISTA study and data is poorly written, almost implying that VISTA is conducted for this study alone. See McCarthy, L., Delbosc, A., Currie, G., Molloy, A.: Parenthood and cars: a weakening relationship? Transportation 4, 1–19 (2018) for some tips on how to better introduce VISTA.
Response: To avoid confusion, we have edited lines 92-94 to give more details about VISTA:
“The sample for this cross sectional study was selected from the Victorian Survey of Travel and Activity (VISTA) 2012-2016, a cross-sectional household travel survey of Greater Melbourne and Geelong in the state of Victoria, Australia, described elsewhere [33]. VISTA runs continuously throughout the year surveying different households each day.”
Comment 8: In the context of the preceding paragraph describing mode to school, the description of the active travel per day needs to be clarified as taken from the respondent’s entire day of transport rather than just the mode to school, assuming that it is.
We have updated lines 86-89 as follows:
“This study aims to investigate if the public transport accessibility of the home and school neighbourhoods are associated with a) walking/cycling or taking public transport to school and b) accumulating 20 minutes or more of daily active travel (to school and/or other destinations) in adolescents aged 12 to 18 years”.
Comment 9: Page 7, there are two rouge sentences: "Adjusted for sex, age, household weekly income. Household size, vehicle ownership, neighbourhood socioeconomic status, distance between home and school, household walkability and school walkability" - not sure where they belong.
Response: Thank you – these comprise a footnote to Tables 2 and 3 and have been moved to the correct position.

Reviewer 2 Report
This paper studies the association between public transport accessibility round adolescents’ home/school and their physical activities, which are important for health benefits. The research is based on analysis of surveys, from which statistical conclusion are made regarding the positive relation between high public transportation accessibility and increased physical activities among adolescents. This work is useful for arranging public transport infrastructure and service for city planners.
The reviewer has the following concerns:
- There are many statistical terms undefined which reduces the readability of the paper. E.g., the statistics of CI and OR have no definition and explanation. Readers have to guess that they are confidence interval and odd rate? Authors should give the mathematical equation of the two values and explain why they are meaningful. Also, what is an “active travel”, which has been coined and mentioned throughout the paper? For maximizing the audience of the paper, the authors should consider readers who are not in the field of statistics and healthcare.
- In section 3, it says “Nearly 64% has their travel diaries completed by proxy”. It is skeptical of whether the reports from proxy other than self report are accurate. An explanation and argument should be made here.
- It is hard to localize the definition of unadjusted and adjusted association. To make it clear, a figure or explanation table for the two terms has to be made.
Author Response
Comment 1: This paper studies the association between public transport accessibility round adolescents’ home/school and their physical activities, which are important for health benefits. The research is based on analysis of surveys, from which statistical conclusion are made regarding the positive relation between high public transportation accessibility and increased physical activities among adolescents. This work is useful for arranging public transport infrastructure and service for city planners.
Response: Thank you for acknowledging our contribution.
The reviewer has the following concerns:
Comment 2: There are many statistical terms undefined which reduces the readability of the paper. E.g., the statistics of CI and OR have no definition and explanation. Readers have to guess that they are confidence interval and odd rate? Authors should give the mathematical equation of the two values and explain why they are meaningful.
Response: We have now added following text to the statistical analysis in methods to explain odds ratio:
Odds ratios (OR) with 95% confidence intervals (CI) comparing two categories (active travel and public transport) with a reference group (motorised vehicle use) were estimated using this modelling approach to quantify the strength of association and its uncertainty.
Comment 3: Also, what is an “active travel”, which has been coined and mentioned throughout the paper? For maximizing the audience of the paper, the authors should consider readers who are not in the field of statistics and healthcare.
Response: We have now described active travel as “walking/cycling for transport” at lines 24 (abstract) and 54.
Comment 4: In section 3, it says “Nearly 64% has their travel diaries completed by proxy”. It is skeptical of whether the reports from proxy other than self report are accurate. An explanation and argument should be made here.
Response: We have recognised this as a study limitation (lines 289-290):
“Secondly, household travel survey data reported on a single day may not reflect habitual travel behaviour and may be prone to inaccuracies in reporting.”
Comment 5: It is hard to localize the definition of unadjusted and adjusted association. To make it clear, a figure or explanation table for the two terms has to be made.
Response: We have added the following text to Statistical analysis to address this concern:
Estimates obtained from regression models adjusted for covariates account for key confounding factors (sex, age, household weekly income, household size, vehicle ownership, neighbourhood socioeconomic status, distance between home and school, household walkability and school walkability) in the relationships between public transport accessibility and active travel outcomes than unadjusted models.
Also see our response to Reviewer 1 comment 9.
